# Effects of Synbiotic Preparation Containing *Lactobacillus gasseri* BNR17 on Body Fat in Obese Dogs: A Pilot Study

**DOI:** 10.3390/ani12050642

**Published:** 2022-03-03

**Authors:** Han-Joon Lee, Jae Hyoung Cho, Woo-Jae Cho, Seong-Ho Gang, Seung-Hwan Park, Bong-Jun Jung, Hyeun Bum Kim, Kun Ho Song

**Affiliations:** 1College of Veterinary Medicine, Chungnam National University, 99 Daehak-ro, Yuseong-gu, Daejeon 34134, Korea; hanjoon1102@cnu.ac.kr; 2Department of Animal Resources Science, Dankook University, Cheonan 31116, Korea; jhcho5216@daum.net; 3Veterinary Nutrition Laboratory, JEIL Feed Co., Ltd., Daejeon 34029, Korea; puco113@naver.com; 4SNH Biotech Co., Ltd., Daejeon 34015, Korea; shgang@snhbiotech.com (S.-H.G.); shpark@snhbiotech.com (S.-H.P.); 5Acebiome Co., Ltd., Seoul 06164, Korea; bjjung@acebiome.com

**Keywords:** obesity, body fat, *Lactobacillus gasseri* BNR17, radiography, computed tomography, microbiome

## Abstract

**Simple Summary:**

The present study evaluated the effectiveness of a synbiotic preparation containing *L. gasseri* BNR17 on body fat in obese dogs. Body weight and the body condition score showed a significant reduction after 10 weeks of synbiotic preparation administration compared with those at baseline (0-week). The subcutaneous fat mass at the level of the third lumbar vertebra and its proportion, compared with the body mass, reduced significantly after 10 weeks of synbiotic preparation administration, compared with those at baseline. Following synbiotic supplementation, microbiome analysis revealed increased microbial diversity, and functional analysis of the microbiota showed an increased abundance of carbohydrate and lipid metabolism. Based on these results, we consider that synbiotic preparation containing *L. gasseri* BNR17 may play a role in reducing body fat and resolving obesity.

**Abstract:**

Obesity is an important health concern in humans and dogs. It can cause a variety of secondary problems, including low bacterial diversity. Several approaches have been tried to solve this problem; one of them is probiotic supplementation. *Lactobacillus gasseri* BNR17 is derived from breast milk and has been proven to be effective for obesity in humans. However, there have been no studies using a synbiotic preparation containing *L. gasseri* BNR17 for obesity management in dogs. Therefore, the present study evaluated the effectiveness of a synbiotic preparation containing *L. gasseri* BNR17 in reducing body fat in obese dogs. A group of obese dogs were fed a synbiotic preparation for 10 weeks. Obesity variables included body weight, body condition score, subcutaneous fat thickness, subcutaneous fat mass and proportion of the fat mass. In addition, feces collected at 0-week and 10-week time points were analyzed for the intestinal microbiome. Results showed a significant decrease in body weight, body condition score, and subcutaneous fat mass and proportion at the level of the third lumbar vertebra. Diversity and functional analysis of the microbiota in obese dogs showed increased microbial diversity, and increased abundance of metabolism of carbohydrate, and lipid after supplementation with a synbiotic preparation. This study was conducted as a pilot study, and the results demonstrated that a synbiotic preparation containing *L. gasseri* BNR17 may play a role in reducing body fat and resolving the obesity in dogs.

## 1. Introduction

Obesity is a growing global health issue in dogs and is found in 34–63% of the population, depending on the location [1,2]. Obesity is defined as chronic excessive body weight causing health problems [1,3]. It results from excessive fat accumulation from high energy intake with low expenditure along with complex interactions of numerous factors including genetic, metabolic, behavioral and nutritional factors [1,3,4,5].

The body condition score (BCS) is a useful evaluation tool that is routinely used to evaluate obesity and body fat distribution in dogs [6,7]. A 9-point scale BCS is usually used; the scoring is performed by visual observation and by palpating the dog [7,8]. Abdominal radiography and computed tomography (CT) can also be used to evaluate body fat distribution in humans and dogs [9,10,11,12]. Subcutaneous fat thickness (ST) can be measured using a lateral abdominal radiograph [10]. For computed tomography, body area (BA), total fat area (TA), and visceral fat area (VA) are measured, and the subcutaneous fat area (SA) is calculated by subtracting VA from TA [10]. Abdominal radiography and computed tomography are well correlated with BCS [10].

The consequences of obesity can be diverse, causing various abnormalities such as chronic inflammatory conditions, increased risk of diabetes mellitus, hypertension, degenerative orthopedic disease, and impaired cardiopulmonary function [1,2,3,13]. Obesity also affects gastrointestinal health by reducing the diversity of the intestinal microbiome [2]. Dogs with obesity have lower levels of serotonin (5-hydroxytryptamine[5HT]) which acts as a neurotransmitter in response to food in the lumen [3,14]. Lower serotonin levels can induce slower colonic transit, which may be followed by an increase in the time taken for fermentation of undigested carbohydrates and lipids and alterations in the microbiome of the intestines [2]. There is a significant difference in microbiomes between obese and lean animals and obese animals have shown a lower bacterial diversity [14,15].

As the awareness of obesity has increased, efforts to treat obesity as a disease have also increased. In human medicine, solutions for obesity are under constant development; these include therapeutic treatments, surgical interventions, and dietary management plans [5,13]. However, these approaches are often associated with side effects and remain ineffective without permanent change to eating and exercise habits [5]. Therefore, there is an increasing interest in safe and effective dietary supplements for body fat management [5].

*Lactobacillus* is a well-known probiotic microorganism and a critical constituent of the intestinal microbiome that is beneficial to human health [5,16,17]. Studies conducted in humans have reported the effect of *Lactobacillus gasseri* BNR17, a probiotic strain isolated from breast milk of healthy females [17]. This strain is characterized by acid and bile resistance, binding to colonic cells, antibacterial properties against food-borne pathogens (*Staphylococcus aureus*, *Escherichia coli* O157:H7, *Listeria monocytogenes*, *Salmonella typhimurium*, and *Bacillus cereus*), and production of antibacterial substances such as bacteriocin [17]. *L. gasseri* BNR17 was found to be effective against obesity and body fat in mice, rats and humans despite there being no change in diet and exercise habits [5,16,17].

However, there have been no studies using a synbiotic preparation containing *L. gasseri* BNR17 for obesity management in dogs. Since *L. gasseri BNR17* showed its potential effectiveness in reducing fat in other species including human, it may provide an efficient way of managing obesity in dogs. The aim of this study was to determine the effectiveness of a synbiotic preparation containing *L. gasseri* BNR17 on body fat in naturally obese dogs.

## 2. Materials and Methods

### 2.1. Ethics Statement

Blood sample collection for the screening test and anesthesia for computed tomography were performed for dogs owned by private individuals. The experimental protocol was approved by the Laboratory Animals Committee of Chungnam National University (approved No. 202003-CNU-024), and the investigators adhered to the Guide for the Care and Use of Laboratory Animals of Chungnam National University.

### 2.2. Animals

This study was designed as a single-center, non-randomized, open clinical trial to examine the effect of *L. gasseri* BNR17 synbiotics on body fat in obese dogs. A total of five privately-owned obese dogs were recruited to the Veterinary Medicine Teaching Hospital of Chungnam National University. Clients were informed of the aim and protocol of the experiment on the first visit, and written informed consent was obtained from each client. The dogs were fasted for 12 hours before the screening tests. The screening tests included history taking, physical examination, complete blood count (CBC), electrolyte analyses, serum biochemistry, total thyroxine (tT4), and adrenocorticotropic hormone (ACTH) stimulation test. These tests were intended to examine the general health condition of the subjects and to exclude subjects with underlying diseases, and all test results were within normal range. Therefore, all subjects were regarded as clear of diseases that are associated with obesity. The specific data of the dogs including age, sex, and breed, are presented in Table 1.

### 2.3. Test Substance

The synbiotic preparation used for this experiment was composed of over 2 × 10^9^ CFU/g of *L. gasseri* BNR17, over 1 × 10^9^ CFU/g of *Lactobacillus plantarum* and other filler powder including non-digestible maltodextrin, galacto-oligosaccharides (GOS), fructo-oligosaccharides (FOS), and polydextrose, and was produced by SNH Biotech Co., Ltd. (Daejeon, South Korea) (Table 2). The pet owners were instructed to administer 1 g of the manufactured product following the provided instructions and not to change the diet of the dogs. 

### 2.4. Body Weight and BCS

Each obese dog had body weight measured on the first day (hereafter referred to as the 0-week time point) and 10 weeks after initiating synbiotic preparation containing *L. gasseri* BNR17 supplementation (hereafter referred to as the 10-week time point). The same digital weighing scale (DB-150, CAS, Gyeonggi-do, South Korea) was used for every measurement. BCS was measured using the 9-point scale at the 0-week and 10-week time points by the same investigator based on illustrations and descriptions provided by the World Small Animal Veterinary Association.

### 2.5. Abdominal Radiography

Each obese dog had an abdominal radiograph with a right lateral view at the 0-week and 10-week time point. A digital system was used to obtain radiographs (MDXP-40TG, Medien International Co., Gyeonggi-do, South Korea). The subcutaneous fat of each dog was measured using a radiograph viewing software (ZeTTa PACS Viewer, TY Soft, Gyeonggi-do, South Korea). Subcutaneous fat thickness (ST) was measured at the level of the third and sixth lumbar vertebrae on a lateral abdominal radiograph for each subject by drawing a straight line from the highest level of the spinous process of the vertebra to the skin–air interface [10]. To account for variations in body size, the ratio of ST to the length of the midbody of sixth vertebrae (rST3, rST6) was calculated [10]. All measurements were performed by the same investigator.

### 2.6. Computed Tomography

Each obese dog underwent a CT scan at the 0-week and 10-week time point. Images were obtained with a CT scanner (Alexion Advance Edition Model TSX-034A, TOSHIBA, Tokyo, Japan), and the software used for viewing the CT images was identical to that used for radiography. On the transverse plane, the region of interest was drawn manually at the level of the third and sixth lumbar vertebrae around the body for body area (BA), total fat area (TA), and surrounding the peritoneal cavity for visceral fat area (VA); the subcutaneous fat area (SA) was calculated by subtracting VA from TA [10]. The ratios of BA to SA (proportion of SA or pSA) were also calculated [10]. All measurements were performed by the same investigator.

### 2.7. Sample Collection and Total DNA Extraction

A total of 10 fecal samples were collected directly from the rectums of the five obese dogs at the 0-week and 10-week time point. By using the QIAamp Fast DNA Stool Mini Kit (QIAGEN, Hilden, Germany), the total DNA was extracted from a total of 200 mg of each dog fecal sample. With the 2 cycles using taco™ Prep Bead Beater (GeneReach, Taichung, Taiwan), the total DNA from fecal samples were extracted by bead-beating, including heating in a water bath (5 min incubation at 70 °C between beatings). For measuring the concentrations of DNA, the Colibri Microvolume Spectrometer (Titertek Berthold, Pforzheim, Germany) was used with OD260/280 ratios of 1.80–2.15 in each DNA sample.

### 2.8. 16S rRNA Gene PCR

For the amplification of the 16S rRNA genes, V3 (5′- TCGTCGG-CAGCGTCAGATGTGTATAAGAGACAGCC-TACGGGNGGCWGCAG-3’) to V4 (5′-GTCTCGTGGGCTCGGAGATGTGTATAAGAGACAGGACTACHVGGGTATCTAATCC-3′) hypervariable regions were used as the PCR primers. Total reaction volume of the PCR amplification was set as 25 μL, containing 1 μM of each primer, 12.5 ng of DNA, and 2× KAPA HiFI HotStart ReadyMix (Roche, Schaffhausen, Switzerland). The PCR conditions were 3 min of initial denaturation at 95 °C, 25 cycles of 95 °C for 30 s, 55 °C for 30 s as annealing, and 72 °C for 30 s of extension, and a final 5 min extension at 72 °C. AMPure XP (Beckman Coulter, Brea, CA, USA) was used to conduct PCR purification.

### 2.9. 16S rRNA Gene Library Preparation and MiSeq Sequencing

To sequence the 16S rRNA gene amplicons, the Illumina MiSeq platform was used through the following steps. First, the random fragmentation of the DNA samples which was followed by 5′ and 3′ adapter ligation, was conducted to prepare the sequencing library. When preparing the library, dual indices and Illumina sequencing adapters were attached to the 16S rRNA gene amplicons, using the Nextera XT Index Kit (Illumina, San Diego, CA, USA). On the final step, the products were normalized and pooled to measure the concentrations of DNA using PicroGreen (Turner BioSystems, Inc., Sunnyvale, CA, USA); the verification of the library size was conducted using the TapeStation DNA ScreenTape D1000 (Agilent Technologies, Inc., Santa Clara, CA, USA). Conditions of PCR cycling were as follows: 3 min initial denaturation at 95 °C, 8 amplification cycles (95 °C for 30 s, 55 °C for 30 s, and 72 °C for 30 s, and a 5 min final elongation at 72 °C.

### 2.10. 16S rRNA Gene Analysis

Once the raw sequence data were generated from the Illumina MiSeq platform, the quality control was performed. rRNA gene data with a length of less than 200 bp sequences and calls containing ambiguous bases were eliminated from the demultiplexed sequence reads to minimize the effects of random sequencing errors. The UCHIME algorithm in Mothur was used to identify chimeric sequences, which were excluded from downstream analysis. Then, the operational taxonomic unit (OTU) picking was conducted using the SortMeRNA and SUMACLUST employed by QIIME (Quantitative Insights into Microbial Ecology) pipeline (version 1.9.1) [18]. Taxonomic assignment of sequence reads was conducted by the naïve Bayesian Ribosomal Database Project (RDP) classifier based on the GreenGenes taxonomy reference database version 13_8. For the downstream analysis, low abundance OTUs and singletons were eliminated from the OTU table with a minimum count of 4 and a low count filter based on a 20% prevalence in the samples. To conduct the data normalization, data were rarefied to the minimum library size. To address the sparsity of the data and the variability in the sampling depth, total sum scaling (TSS) was used before any statistical comparison. Based on the assigned taxa of the OTU, the investigation of the relative abundance of functional categories was conducted using PICRUSt, which was used to interpret the 16S amplicon sequencing data in terms of biological pathways [19]. PICRUSt was utilized to predict the abundance of gene categories (COGs) and metabolic pathways (KEGG). For visualization of investigated gene functions, GraphPad Prism v7.00 (GraphPad Software, San Diego, CA, USA) for relative abundance was used.

### 2.11. Statistical Analysis

Statistical analysis was performed using a commercial software program (IBM SPSS Statistics 23.0, SPSS Inc., Chicago, IL, USA). Data for rST3, rST6, SA3, pSA3, and SA6 were normally distributed and represented as means with standard error of the means; data for the 0-week and 10-week time points were compared using a paired t-test. Body weight, BCS, and pSA6 were not distributed normally and were represented as the median; data for the 0-week and 10-week time points were compared using the Wilcoxon signed-rank test. For statistical microbiome analysis, GraphPad Prism v7.00 (La Jolla, CA, USA) was used to visualize the data. The GraphPad Prism v7.00 was used to compute alpha diversity indices, including Observed OTUs, Chao1, Shannon, and Simpson. A non-parametric Kruskal–Wallis test was conducted to compare the alpha diversity among the groups. Principal coordinate analysis (PCoA) plots at the OTU level at 97% identity were generated using weighted and unweighted UniFrac distances. Analysis of similarities (ANOSIM) based on the unweighted and weighted UniFrac distances was performed to compare the beta diversity among the groups. A *p* value < 0.05 was considered to be statistically significant.

## 3. Results

### 3.1. Differences in Body Fat Status before and after Synbiotic Preparation Containing L. Gasseri BNR17 Administration

Differences in body fat status before and after synbiotic preparation containing *L. gasseri* BNR17 administration were analyzed (Table 3). Results showed significant effects on body weight, BCS, SA3, and pSA3. The median of body weight declined significantly from 14.0 kg at the 0-week time point to 13.5 kg at the 10-week time point (*p* < 0.05). The median BCS significantly decreased from 9.0 at the 0-week time point to 7.5 at the 10-week time point (*p* < 0.05). In addition, significant decreases were observed in the mean SA at the level of the third lumbar vertebra from 99.86 ± 15.57 cm^2^ at the 0-week time point to 82.70 ± 13.92 cm^2^ at the 10-week time point (*p* < 0.05), and in the proportion of SA at the level of the third lumbar vertebra from 36.14 ± 2.44 % at the 0-week time point to 31.05 ± 3.24 % at the 10-week time point (*p* < 0.05) (Table 3). Other variables, including rST3, rST6, SA6, and pSA6, showed no significant differences.

### 3.2. Alpha Diversity Analysis of the Microbiota in Obese Dogs

In the intestinal microbial diversity analysis, the alpha diversity (observed OTUs, Chao1, Shannon, and Simpson) analysis of obese dogs following synbiotic preparation containing *L. gasseri* BNR17 supplementation confirmed that the level of species diversity showed increasing propensity of normalized abundance (Table 4). 

### 3.3. Beta Diversity Analysis of the Microbiota in Obese Dogs

Beta diversity (PCoA) analysis confirmed that the microbiota composition showed significantly different clustering before and after synbiotic preparation containing *L. gasseri* BNR17 supplementation (Figure 1). At the phylum level, the bacterial sequences from the 0-week samples were composed predominantly of the phyla *Firmicutes* (80.07%), *Proteobacteria* (13.48%), *Bacteroidetes* (3.93%) and other phyla of the total sequences analyzed (Figure 2A). The 10-week samples consisted largely of the phyla *Firmicutes* (59.41%), *Bacteroidetes* (18.34%), *Fusobacteria* (10.73%) and other phyla of the total sequences analyzed (Figure 2A). At the genus level, *Enterococcus* was the most enriched genus in the 0-week samples (48.82%; Figure 2B), whereas *Blautia* was the most enriched genus in the 10-week samples (24.35%; Figure 2B). The relative abundance of *Enterococcus* decreased significantly from an average of 48.82% in the 0-week samples to 10.98% in the 10-week samples (Figure 3A). The relative abundance of *Bacteriodes*, *Collinsella* and *Prevotella* showed increased propensity from 6.81%, 0.90%, and 0.00% in the 0-week samples to 21.58%, 6.59%, 3.21% in the 10-week samples, respectively (Figure 3B–D).

### 3.4. Functional Analysis of the Microbiota in Obese Dogs

Based on the KEGG pathway analysis, the functions of microbes present in the samples with synbiotic preparation containing *L. gasseri* BNR17 supplementation are shown in Figure 4. Even though there were no significant differences, the metabolism of carbohydrates and lipids were enriched after administration.

## 4. Discussion

This study was conducted to explore the effect of a synbiotic preparation containing *L. gasseri* BNR17 on body fat in obese dogs. There are many strains of *Lactobacillus*, but *L. gasseri* BNR17 has shown promising results in body fat reduction in other species and has not been previously used in dogs. 

Kang et al. [16] suggested that *L. gasseri* BNR17 administration reduced body weight gain in mice with high-sucrose diet-induced obesity. Another study of Kang et al. [17] also reported that the percent increase in weight and adipose tissue was significantly lower in rats with diet-induced obesity following treatment with *L. gasseri* BNR17 than in the control. In humans, Jung et al. [5] reported body weight reduction in the obese group after 12 weeks of *L. gasseri* BNR17 administration and little change in the placebo group; however, these results did not attain statistical significance. Our results were consistent with these findings; there was a significant decrease in body weight in dogs after synbiotic preparation containing *L. gasseri* BNR17 administration compared with that at baseline.

In addition, Jung et al. [5] revealed that there was a significant decrease in body mass index from the 0-week time point to the 12-week time point in the *L. gasseri* BNR17-administered group of human subjects with obesity compared with that in the placebo group. We used BCS, a standard measure of obesity in dogs, and found a significant decrease in BCS after 10 weeks of synbiotic preparation containing *L. gasseri* BNR17 supplementation.

Body fat can be assessed by measuring ST and calculating the ratio against the length of the midbody of sixth lumbar vertebrae in lateral abdominal radiographs [10]. In this study, the ratios of ST to the length of the vertebral midbody of sixth lumbar vertebrae were compared between the 0-week time point and 10 weeks after synbiotic preparation containing *L. gasseri* BNR17 administration; we found no significant difference. Kim et al. [10] noted that when measuring ST, some muscles may be included, the amount of which may vary due to sex or breed. This can influence the accuracy of subcutaneous fat measurement, even when adjusted with a calculated ratio to the vertebral midbody. This is a possible reason for the insignificance of these results. Therefore, additional studies on subcutaneous fat measurement in abdominal radiographs, while considering the variations due to sex and breed are needed.

A significant reduction in subcutaneous fat along with mesenteric and epididymal fat volume was observed in CT scans of obese mice after *L. gasseri* BNR17 synbiotics administration [16]. Additionally, there was a significant decrease in waist and hip circumferences measured using a measuring tape 12 weeks after supplementation of *L. gasseri* BNR17 synbiotics, although CT scans were not performed in humans [5]. In the present study, the subcutaneous fat mass and proportion of it showed significant reduction when measured at the level of the third lumbar vertebra, but not at the level of the sixth lumbar vertebra. In humans, CT scans are performed at the level of the umbilicus, which corresponds to the third lumbar vertebra level in dogs [9,12]. This correspondence may explain why the measured data at the third lumbar vertebra showed significance. In contrast, data from the sixth lumbar vertebra had no significant difference due to variations in fat distribution, depending on the breed and characteristics of the dog [11]. Additional studies measuring body fat mass via CT scans in various dog breeds are needed. While there is a more reliable method to assess body composition, including fat volume, via intravenous administration of deuterium oxide dilution followed by CT scan, this method is not clinically practical, as it requires considerable time and resources [20]. New methods are required for measuring body fat volume with more efficiency.

Microbiome analysis showed that supplementation with a synbiotic preparation containing *L. gasseri* BNR17 in obese dogs altered the predominant phyla (0-week: *Firmicutes*, *Proteobacteria*, and *Bacteroidetes*; 10-week: *Firmicutes*, *Bacteroidetes*, and *Fusobacteria*). The abundance of *Firmicutes* and *Proteobacteria* declined, whereas that of *Bacteroidetes* and *Fusobacteria* increased. The composition of the microbiota at the genus level was also modified. There was a decrease in the abundance of *Enterococcus* and an increase in the relative abundance of *Bacteroides*, *Collinsella*, and *Prevotella* in the 10-week group compared with that in the 0-week group. The genus *Enterococcus* is the leading cause of bacteremia, endocarditis, and urinary tract infections. Bacteremia is often preceded by dense colonization of the GI tract, from which *Enterococcus* can translocate into the bloodstream [21]. The decrease in the abundance of *Enterococcus* following synbiotic preparation containing *L. gasseri* BNR17 supplementation suggests promising effects of the approach against these infections. However, *Enterococcus* is also known to play a role as a beneficial probiotic (e.g., *E. faecium*, *E. faecalis*). Therefore, further studies will help us to understand the exact roles of *Enterococcus* in dogs in association with *L. gasseri* BNR17. *Bacteroides*, *Collinsella*, and *Prevotella*, which are known to ferment carbohydrates, producing volatile fatty acids that are reabsorbed through the large intestine and utilized by the host as an energy source [22], are correlated positively with circulating insulin [23], and reduce intestinal inflammation [24]. In the present study, an increase in the abundance of these genera in the 10-week group suggests the efficacy of the synbiotic preparation containing *L. gasseri* BNR17 in promoting beneficial bacteria, which may eventually contribute to improved gut performance and health.

Our KEGG pathway analysis showed that metabolism of carbohydrates and lipids was enriched after the administration of the synbiotic preparation containing *L. gasseri* BNR17. Pathways including “Fructose and mannose metabolism”, “Starch and sucrose metabolism”, “Pentose phosphate pathway”, “Citrate cycle (TCA cycle)”, “Fatty acid biosynthesis”, and “Biosynthesis of unsaturated fatty acids”, contribute to glycolysis, resulting in production of ATPs, which helps to improve various metabolisms by providing energy to living cells [25,26]. As ATP increases, the energy is used to generate heat, resulting in dissipation of calories and fat loss [27]. “Butanoate (butyrate) metabolism” contributes to short-chain fatty acid (SCFA) production [25], which can be closely correlated with the increase in abundance of *Prevotella*, which helps to produce SCFAs. Both “Primary and Secondary bile acid biosynthesis” may help obese groups to facilitate digestion of dietary fats and oils [24,25,28]. In conclusion, following the consumption of a synbiotic preparation containing *L. gasseri* BNR17, the increase in carbohydrate and lipid metabolism was beneficial to the microbiome of dogs. In addition, studies have shown that the gut microbiome has an effect on nutrient metabolism and energy expenditure. Treatment of obese people with probiotics induced microbial shifts including the diversity and composition of the gut microbiome [29,30]. As such the results observed from this study were similar to those of previous human studies, which indicates that the gut microbiome has a strong relationship with obesity. Nevertheless, further studies, such as shotgun metagenomics sequencing, will be helpful to understand the exact mechanisms of obesity-related functions of the microbiome.

There are other components including *L. plantarum* and prebiotics such as GOS and FOS that contribute to intestinal health. *L. plantarum* plays an important role in enhancing the innate immune system with antimicrobial activity and other properties [31]. Prebiotics such as GOS and FOS are metabolized by *Lactobacillus*, which is one of the beneficial bacteria, by decreasing the intestinal pathogens and providing health-related bacterial metabolites [32]. The effects of increasing the growth of *Lactobacillus* are intensely debated. There are several studies about the effects of different concentrations of FOS on *Lactobacillus*. However, according to Chen et al. [33], FOS with proper concentration may help increase the number of *Lactobacillus*. Endo et al. [34] showed utilization of FOS by *Lactobacillus* and Sako et al. [35] showed utilization of GOS by *Lactobacillus* resulted in increased enzyme activity. There could be diverse mechanisms of prebiotics depending on oligosaccharides, so optimal combinations of prebiotics with *L. gasseri* BNR17 need to be studied.

This study had two limitations. First, a small number of dogs, diverse in species, gender, and age, were utilized. Second, the probiotic was fed over a relatively short period of time.

## 5. Conclusions

Body weight, BCS and SA at the level of the third lumbar vertebra and its proportion showed significant reduction after 10 weeks of synbiotic preparation containing *L. gasseri* BNR17 administration compared with those at baseline. Diversity and functional analysis of the microbiota in obese dogs showed increased microbial diversity, and increased abundance of metabolism of carbohydrate and lipid. This pilot study indicates that feeding a synbiotic preparation containing *L. gasseri* BNR17 may play a role in reducing body fat and resolving obesity in dogs.

## Figures and Tables

**Figure 1 animals-12-00642-f001:**
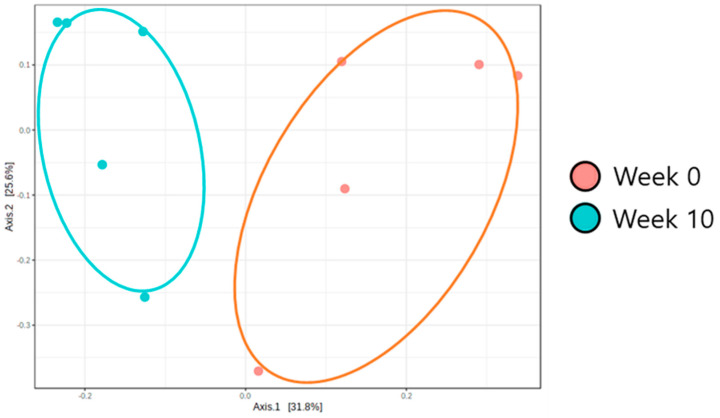
*L. gasseri* BNR17 intestinal microbiome beta diversity analysis according to synbiotic prep-aration containing *L. gasseri* BNR17 supplementation (PCoA: Principle Coordinates Analysis, un-weighted UniFrac Distance) (R: 0.552, *p* < 0.01).

**Figure 2 animals-12-00642-f002:**
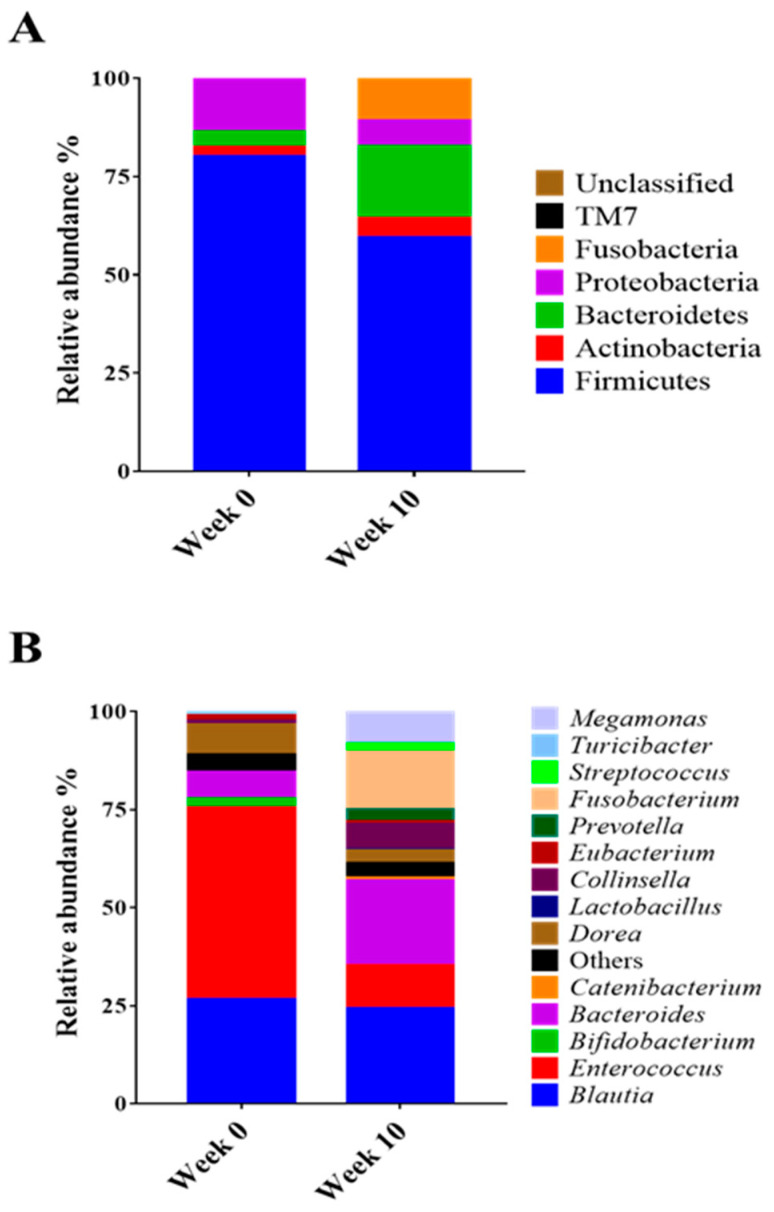
Comparison of taxonomic composition between obese dogs according to synbiotic preparation containing *L. gasseri* BNR17 supplementation; (**A**): Phylum, (**B**): Genus.

**Figure 3 animals-12-00642-f003:**
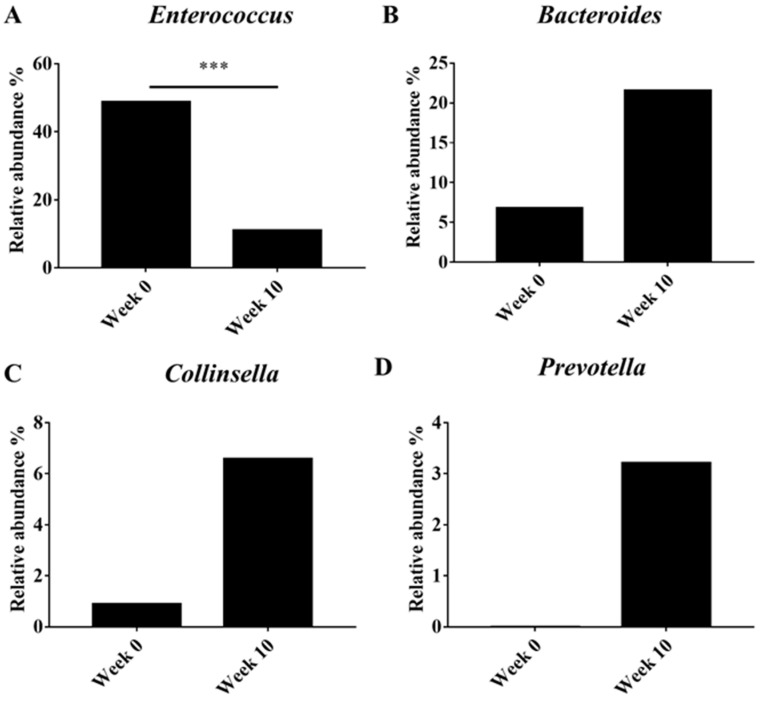
Comparison of microbiota between obese dogs according to synbiotic preparation containing *L. gasseri* BNR17 supplementation. Genus: (**A**) *Enterococcus*, (**B**) *Bacteriodes*, (**C**) *Collinsella*, (**D**) *Prevotella*. *** *p* < 0.001.

**Figure 4 animals-12-00642-f004:**
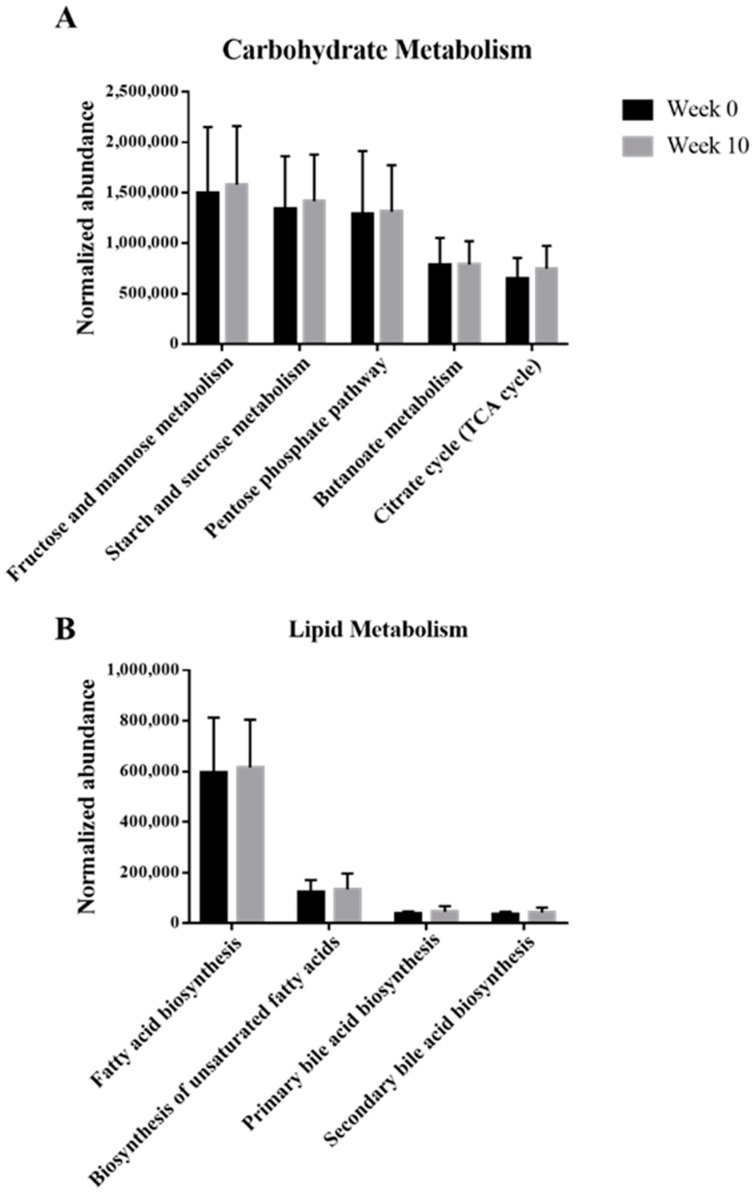
Functional annotation of the gut microbiomes between week 0 and 10 with the administration of the synbiotic preparation containing *L. gasseri* BNR17. Normalized abundance of the level 2 KEGG subsystem classified reads associated with the metabolism of carbohydrate (**A**), and lipid (**B**). The error bars show the calculated standard error of the means of five replicates of the samples.

**Table 1 animals-12-00642-t001:** Age, sex, and breed of the obese dogs in this study.

Group	Age (Year)	Sex	Breed
Obese Dogs	8.59 ± 3.81	Spayed Female (3)	Beagle (3)
		Female (1)	Spitz (1)
		Castrated male (1)	Yorkshire Terrier (1)

**Table 2 animals-12-00642-t002:** Composition and proportions of symbiotic preparation in this study.

Item	Percentage
*L. gasseri* BNR17	6.00%
*L. plantarum*	3.40%
Galacto-oligosaccharides	6.00%
Fructo-oligosaccharides	6.00%
Non-digestible maltodextrin	18.30%
Polydextrose	60.00%
GOX	0.30%
Total	100%

**Table 3 animals-12-00642-t003:** Comparison of obesity variables between 0-week and 10-week.

Variables	Time Point		t/Z	*p*
	0–Week	10–Week		
Body weight (kg)	14.0 (9.15–21.38)	13.5 (8.60–19.00)	−2.023	0.043 *
Body condition score	9.0 (7.75–9.00)	7.5 (7.00–8.25)	−2.041	0.041 *
Ratio of subcutaneous fat thickness at L3	1.61 ± 0.09	1.51 ± 0.08	1.466	0.216
Ratio of subcutaneous fat thickness at L6	2.46 ± 0.25	2.33 ± 0.17	1.569	0.192
Subcutaneous fat area at L3 (cm^2^)	99.86 ± 15.57	82.70 ± 13.92	3.244	0.032 *
Proportion of subcutaneous fat area at L3 (%)	36.14 ± 2.44	31.05 ± 3.24	4.355	0.012 *
Subcutaneous fat area at L6 (cm^2^)	130.91 ± 25.98	111.27 ± 18.20	1.612	0.182
Proportion of subcutaneous fat area at L6 (%)	48.03 (45.75–59.64)	44.10 (40.15−53.33)	−1.753	0.080

* *p* < 0.05, Normally distributed data are presented as the mean ± SEM. Data that were not normally distributed are presented as the median.

**Table 4 animals-12-00642-t004:** Analysis of changes in intestinal microbial diversity in obese dogs according to synbiotic preparation containing *L. gasseri* BNR17 supplementation.

Alpha Diversity	Obesity		*p*
	0–Week	10–Week	
Observed OTUs	824.40 ± 176.07	1202.20 ± 141.07	0.52
Chao1	3155.85 ± 1062.21	3397.39 ± 431.51	0.68
Shannon	2.97 ± 0.49	3.92 ± 0.35	0.99
Simpson	0.71 ± 0.12	0.86 ± 0.04	0.99

Observed OTUs: number of species distributed per sample. Chao1: estimates the richness of the species based on the information of the species. Shannon: estimates the diversity of the species present in the sample. Simpson: the concentration of the species found in the sample, which has a value of 0 to 1.

## Data Availability

The data presented in this study are available in this paper.

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
