# Peer review of "Effects of Synbiotic Preparation Containing Lactobacillus gasseri BNR17 on Body Fat in Obese Dogs: A Pilot Study"

_animals, 2022, doi:10.3390/ani12050642_

Round 1
Reviewer 1 Report
See attached word document for comments on this manuscript.

Author Response
We thank you for your thorough review and comments.
Please see the attachment.

Reviewer 2 Report
This is a very interesting topic, but the introduction fails in justifying why Lactobacillus gasseri, could be an accurate solution to treat obesity. The number of dogs, enrolled in this study, is quite low and the population looks quite unusual. When we compare the breeds, mentioned in Table 1 and the initial weights in Table 3, we face some surprise. It is well understood that we treat obese dogs, but a Yorkshire Terrier of 8.15 kg is a very severe case. A Table with individual data would have been preferable. It is more a caseload, than a controlled study, and such work gets significant improvement with sharing individual data and evolution. By the way, a description of the diet would have been interesting and could provide valuable data for the discussion about the evolution of the microbiome. The statistics are very poorly detailed, especially with explaining whet is compared (I guess it is the difference between week 0 and week10). The discussion does not provide any clarification about the justification of the treatment. Since the idea comes from human medicine, it would have been interesting to compare the evolution in human and dog patients, to discuss the evolution of the weight and microbiome.Author Response
We thank you for your thorough review and comments.
Please see the attachment.

Round 2
Reviewer 1 Report
See attached document with comments

Reviewer 2 Report
Thank you for improving significantly your paper. In my opinion, it would be better to modify the title and present it as a Preliminary or Pilote study.
Author Response
We really thank you for your efforts and comments.
We have changed the title and presented as pilot study.